# Retirement Income and Financial Market Participation in New Zealand †

**Xiaobo Xu** [1] , **Martin Young** [2] , **Liping Zou** [3,*] and **Jiali Fang** [3]

1   School of Economics and Management (School of Tourism), Dalian University, Dalian 116000, China
2   School of Economics and Finance, Massey University, Palmerston North 4442, New Zealand
3   School of Economics and Finance, Massey University, Auckland 0745, New Zealand
*   Correspondence: l.zou@massey.ac.nz
†   Access to the data used in this study was provided by Statistics New Zealand under conditions designed to give effect to the security and confidentiality provisions of the Statistics Act 1975. The results presented in this study are the work of the authors, not Statistics New Zealand.

**Abstract:** Using New Zealand Household Economic Survey (HES) 2018 data, we examine the impact of direct financial market participation post-retirement on retirement income in New Zealand. Our results demonstrate the importance of post-retirement financial market participation in the enhancement of retirees' financial well-being. We conclude that retirees who participate in the financial market enjoy a 78% increase in overall annuitised net wealth; further analysis also reveals a substantial 154% increase if government pensions are excluded from calculations of annuitised net wealth. Moreover, these retiree participants also show higher probabilities of financial-situation satisfaction. These results highlight the significant contribution to retirement income of direct financial market participation. Our paper sheds extra light on issues related to retirement financial well-being and has important implications for policy makers in New Zealand.

**Keywords:** financial market participation; New Zealand retirement income; annuitised net wealth; financial-situation satisfaction; life satisfaction

**JEL Classification:** D14; D31; G11; G51

## 1. Introduction

The pension system plays a significant and vital role in supporting citizens' retirement life worldwide. It should be designed to provide security for retirees to maintain a minimum living standard, especially for those with a government pension as their single source of income. The pension systems in many countries are organised and affected by demographic ageing. The Organization for Economic Co-operation and Development (OCED) reported 17.2% of the population as ageing (aged 65 years and above) across all OCED countries in 2018, with this number expected to rise to 27.1% by 2050.[1] The ongoing ageing process implies less saving among retirees with longer life expectancy and poses significant challenges to the sustainability of the pension system. Therefore, it is necessary for the system to be reviewed and reformed to ensure that it is adequate and sustainable.

New Zealand's three-pillar pension system is one of the world's most generous systems and is considered to be at the forefront for its adequacy and sustainability. The first pillar is New Zealand superannuation (NZ Super); any retiree aged 65 years or above who passes the residency test qualifies for it. The second pillar is auto-enrolled voluntary Kiwi-Saver, with employees and employers making contributions to it. The third pillar is the private saving conducted by retirees themselves in the form of defined contributions and/or defined benefit plans, annuities and other types of savings.[2] However, the NZ Super rate serves the low-income group well if its intended purpose is to keep older people out of poverty, with its primary goal being to provide social protection rather than

replace earnings. For the high-income group, New Zealand's second and third pillars are not comparable to the arrangements of other countries with mandatory savings schemes and other programmes. New Zealand Ministry of Social Development data show that there were 774,651 people receiving New Zealand Super in March 2019. By 2068, the number is projected to be 1,838,100, representing 28.21% of the population.[3] Moreover, New Zealand fell from 10th to 15th place among 43 countries, according to Mercer's ranking in 2021, mainly due to the significant drop in the adequacy sub-index.[4] This unarguably presents significant social and economic challenges for the New Zealand government. New Zealanders have obviously become increasingly liable to save for their own retirements. In addition, there are voices to increase the pension qualifying age from 65 to 67[5], while New Zealand policy makers are forced to devote significant resources and time to encourage private saving through financial literacy education and direct financial market participation. It has been believed that financial markets are crucial to a nation's economic growth and that financial market participation is the engine needed to drive an individual's net wealth. Thus, in this study, we attempt to answer the following important question: Does post-retirement financial market participation improve retirement income, thereby enhancing retiree objective and subjective financial well-being in New Zealand? The New Zealand Household Economic Survey (HES) 2018 data provide us with a means of examining this important and interesting question in the New Zealand context. We also investigate the extent to which, and through what channels, financial market participation affects retirement income.

Giannetti and Koskinen (2010) document that New Zealand ranks second in domestic stock market participation, behind Australia and ahead of the UK, Japan, Denmark and the US.[6] The New Zealand equity market has experienced steady growth of 11.2%, with a 7.2% return over the 2008 to 2018 period, according to the New Zealand Stock Exchange and the Financial Markets Authority (FMA).[7] The development of the domestic equity market would encourage individuals to hold shares, which is part of financial market participation. Therefore, financial market participation provides an alternative income source for retirees as a special group with limited earning capacity. Typically, retirees are risk-averse and prefer safe assets with low interest rates, such as term deposits. However, this option is not optimal when New Zealand interest rates are low, and retirees are disadvantaged significantly. Moreover, decreasing retirement income and longer life expectancy encourage retirees to seek for growth-like assets to alleviate their economic pressure. Macro-level and micro-level constraints motivate retirees to invest in the financial market post-retirement. Retirees may have indirectly participated in the financial market through the defined contributions of their retirement plans; however, this is a passive strategy and not the focus of this study. In this paper, we focus on direct financial market participation, including the holding of shares, unit trusts and managed funds, where retirees have direct and full control. Thus, if a retiree has invested in equity markets, unit trusts and/or managed funds, they are considered to directly participate in the financial market. The HES 2018 data enable us to identify this information. The HES data also allow us to calculate annuitised net wealth (with and without government pensions) as our objective measure of retirement income. Annuitised net wealth has been used in the existing literature as a way of measuring retirement income, as documented in Burnett et al. (2018) and Haveman et al. (2006). In addition, we use HES survey scores for retirees' overall life satisfaction and financial-situation satisfaction as alternative subjective measures of retirement income.

From a sample of 2175 retirees, we conclude that direct financial market participation post-retirement has a positive and significant impact on retirement income, when retirement income is proxied by both objective and subjective measures. We find that retirees who directly participate in the financial market experience an approximately 78% increase in annuitised wealth compared to those who do not. This rate increases substantially to 154% when the government pension is not included when calculating annuitised net wealth, indicating that direct financial market participation increases private savings, which contributes significantly to post-retirement income. Further investigation of control

variables reveals that older retirees with higher regular gross retirement incomes, better education and home contents insurance are likely to have higher retirement incomes for both objective measures. On the other hand, we find that retirees living with a partner and working retirees suffer from reduced retirement income as measured by annuitised net wealth with government pensions.

We further conduct our analysis using subsamples, including age, gender, partnership-status, living-area and employment-status subsamples. Our results highlight the different influence channels among partnership-status, living-area and employment-status retiree groups; for example, the positive impact of financial market participation on retirement income is reflected in the annuitised wealth channel for working retirees living with a partner and living in other cities. On the other hand, it is reflected in the subjective financial well-being channel for those retirees who are currently not working, living without a partner and living in major urban cities. Younger male retirees have significantly higher financial-market-participation rates, resulting in greater retirement income according to objective and subjective measurements. We use several alternative approaches to conduct robustness checks; all results are consistent with our baseline results, suggesting that our main results are robust.

We contribute to the existing literature in several important ways: 1. we consider the impact of direct financial market participation post-retirement on retirement income, using both objective and subjective measurements from New Zealand HES 2018 survey data. Much of the prior literature used either objective (Haveman et al. 2007b) or subjective measurements (Bonsang and Klein 2012), while we use both; 2. we use annuitised net wealth without government pension as an alternative proxy for objective measurement, which enables us to determine the significant contribution of individual private savings to post-retirement overall wealth (government pensions included) through financial market participation; 3. we only focus on retirees' post-retirement participation, while the prior literature has mostly focused on the entire population (Bilias et al. 2017; Fagereng et al. 2017); and 4. our results have important policy implications, such that New Zealand policy makers should allocate adequate resources to promote financial literacy in order to increase direct financial market participation.

The remainder of the paper is organised as follows. Section 2 briefly discusses the institutional background of the New Zealand retirement system and the relevant literature. Section 3 describes the data and methodology. Section 4 presents the empirical results, and Section 5 concludes.

## 2. Institutional Background and Literature Review

### 2.1. The Ageing Population and the Retirement Pension System in New Zealand

The median age of the New Zealand population increased from 35.9 years old in 2006 to 37.4 years old in 2018, with the ageing population (those aged 65 and above) rising from 12.1% in 2001 to 15.2% in 2018 and being expected to reach 22.22% in 2036. The growth rate for 65-plus individuals is 10.5 times faster than that for those under 14, with an old-age dependency rate of 23.5 for every 100 working people. This ratio reveals that many retirees are supported by young, working generations. For the period from 2017 to 2019, life expectancy was 80 years for males and 83.5 years for females, while it was 67.2 years for males and 71.3 years for females during the period 1950–1952. The increase in life expectancy represents the extension of each age group. However, the gross cost of New Zealand superannuation is projected to increase from 4.8% of GDP in 2015 to 7.2% in 2045 and to reach 7.9% by 2060, according to the NZ Treasury 2017 report[8]. On an after-tax basis, this is equivalent to 4.2% of GDP for 2016, 6.1% for 2040 and 6.7% for 2060 (Davey and Stephens 2018). Thus, the increase in the ageing population will impose significant pressure on the government pension system over time. Therefore, raising the compulsory retirement age may provide an effective way to release the burden on government expenditure on public pensions. Alternatively, an even more effective way to address this issue is to promote private savings through direct financial market participation.

New Zealand has a generous three-pillar pension system. NZ Super is the first pillar, and retirees qualify for NZ Super if they pass the residency test, according to which they must be a citizen, a permanent resident or a residence class visa holder. The NZ Super is a flat-rate pension for all qualified retirees, of around NZD 25,000 a year before tax paid to a single person and NZD 40,000 to a couple in 2021[9], and is treated as taxable income. NZ Super is designed to keep up with increases in costs during retirement, as it increases in line with inflation or average after-tax wages. However, NZ Super cannot guarantee the standard of living of those in relatively higher income brackets before retirement, such that they have to top up their private savings in order to maintain a similar standard of living (Makhlouf 2011). Thus, NZ Super only satisfies those lower-income retirees who made less than 50% of the average wage before retirement. According to the Commission for Financial Capability, New Zealanders hold different views on the role of NZ Super; some consider it a way to meet the basic standard of living, while others see it as a reward for working hard and paying taxes.[10] From both points of view, it is indicated that New Zealanders consider NZ Super to be a minimum standard for retirement living and that retirees may be unable to maintain their desired lifestyles from this single source of income. Moreover, the New Zealand Retirement Expenditure Guidelines (2021) report an increase in retirement spending in excess of NZ Super, and the gap is widening.[11]

KiwiSaver is a defined voluntary contribution scheme, considered the second pillar of the New Zealand retirement system, and was introduced in July 2007.[12] It is a combined contribution from employees and employers, with the minimum contribution rate for an employee being 3% of before-tax income. KiwiSaver is an auto-enrolment scheme, and it functions like compulsory superannuation in its design, as employees rarely withdraw from it. Retirees benefit tremendously from the default option, as people tend to keep the status quo and are reluctant to make changes. Due to the short span of this scheme, however, many retirees (those aged 65 and above) do not benefit from this scheme, and some employees have withdrawn from the scheme despite the low withdrawal rate.[13] Before the introduction of KiwiSaver, the occupational savings scheme (provided both by the government and by the private sector) included around 15% of the workforce, and the enrolment rate has been declining (2015, NZIER report).[14] In addition, KiwiSaver helps to improve the financial situations of those retirees with low to middle levels of wealth but is not significant for high-income retirees, as wealthy retirees tend to have alternative plans to maintain their pre-retirement lifestyles.

The third pillar is voluntary saving schemes. This refers to any type of retirement saving scheme from self-selected defined contributions and/or defined benefit plans, annuities, life insurance and other types of savings. Retirees may choose their own supplementary third-pillar retirement plans before or after retirement to secure sufficient income for retirement life. The third pillar is not compulsory, and it serves as an additional source of retirement income to supplement the first two pillars, providing retirees with their desired retirement lifestyles. However, it is difficult for retirees to maintain their pre-retirement lifestyles without savings from the third pillar. There is a broad range of voluntary retirement pension plans. In this paper, we only focus on direct financial market participation, that is, retirees' direct holding of New Zealand company shares, managed funds and unit trusts. We attempt to examine the role of direct financial market participation post-retirement in enhancing retiree income.

Makhlouf (2011) documents that New Zealand has the lowest poverty rate among ageing populations, while Dang et al. (2006) concluded that, in countries with more support for their ageing populations, higher levels of poverty among retirees may result. However, the low poverty ratios among various countries are the result of complete home ownership among retirees, as most pay off their mortgages upon retirement. This phenomenon implies that the poverty rate is significantly affected by home ownership in the ageing population. Property can be considered an extra pillar to provide another layer of income (Kronick and Laurin 2016), and it is normally considered the last resort (Skinner 2007). Furthermore, Statistics NZ (2020) revealed that the homeownership rate has been declining and that

household income in New Zealand is relatively low among OECD countries.[15] It is therefore time for New Zealanders to focus on private savings for retirement, especially given that life expectancy tends to increase over time.

### 2.2. Review of the Literature on Retirement Income

The objective measurement of retirement income focuses on the value of wealth and is used to calculate the cashflow to be distributed over remaining retirement life. This measure is defined as annuitised net wealth (Haveman et al. 2007a; Haveman et al. 2006), where individual net wealth is annuitised based on expected life expectancy. Annuitised net wealth is a reliable way of gauging retirees' retirement wealth, as life expectancy and risks, as reflected by the discount rate, are taken into account. However, it is difficult to define the threshold for income sufficiency; for example, it is documented that 30% of retirees are unable to meet the 70% pre-retirement income benchmark, with 5% having inadequate resources as measured according to the national poverty line in the US (Haveman et al. 2006). Annuitised net wealth provides a benchmark amount that retirees are expected to receive during retirement; however, it does not provide sufficient information on the sustainability of a retiree's pre-retirement lifestyle after retirement (Bernheim et al. 2001). Therefore, an alternative measure would provide insight into a comprehensive understanding of retirement income. Subjective well-being has been widely applied and considered to be a good proxy for the measurement of utility functions (Finkelstein et al. 2009). Bonsang and Klein (2012) documented that involuntary retirement schemes result in negative impacts on overall satisfaction regarding retirement financial well-being. Elder and Rudolph (1999) concluded that pre-retirement financial literacy education enhances financial well-being in the US. Therefore, we use overall life and financial-situation satisfaction as our subjective measures for retirement income.

Life satisfaction is related to a wide range of factors, such as health, income (Layard et al. 2008) and other personal characteristics, while financial satisfaction is more closely related to wealth position. Therefore, overall life satisfaction and financial-situation satisfaction may exhibit distinct patterns. For example, Alan et al. (2008) reported that life satisfaction appears to fall with age, while financial-situation satisfaction seems to rise with age in Canada. They also documented that financial dissatisfaction is more closely related to involuntary retirement. Barrett and Kecmanovic (2013) derived similar results, showing that subjective well-being either improves or remains unchanged after retirement for the majority of retirees, while deteriorating well-being is largely due to job loss and/or bad health conditions. However, subjective measures may not necessarily be closely related to income or wealth, as they represent only one ingredient of overall life satisfaction. For example, Bond and Lang (2019) documented that subjective measurement may not be as reliable as objective measurement. Nevertheless, subjective life satisfaction and financial well-being measurements may still provide indispensable information respecting retiree income.

### 2.3. Financial Literacy and Financial Market Participation

The Commission for Financial Capability (CFFC) Statement of Intent 2021–2024[16] highlights that 14% to 18% of New Zealand employers are actively planning for an ageing workforce, while 26% of men and 41% of women under 65 are ignorant as to how much money they will need for a comfortable retirement. Thus, there exists an urgency to promote financial literacy and financial market participation, as financial literacy and retirement income are significantly correlated (Fong et al. 2020).

Financial literacy is the ability to understand and use financial skills to manage financial resources effectively for a lifetime of financial well-being (Hung et al. 2009). Three terms are commonly used in designing survey questionnaires to evaluate an individual's level of financial literacy, namely, inflation, compounding and diversification (Bernheim 1998; Gustman and Steinmeier 2005; Lusardi 2006). A number of demographic factors are also considered to be related to financial literacy, for example, gender (Lusardi and Mitchell 2008) and level of education (Worthington 2006). In addition, much evidence has been

documented in international settings (Almenberg and Säve-Söderbergh 2011; Boisclair et al. 2017; Bucher-Koenen and Lusardi 2011; Kalmi and Ruuskanen 2018; Moure 2016). Crossan et al. (2011) revealed that financial literacy is insignificantly associated with retirement planning in New Zealand among those aged 18 years and above; this may be due to New Zealand's universal public pension systems. However, a more recent study (Noviarini et al. 2021) surveying those aged 55 and older argues that financial literacy is important for retirement preparedness.

Financial market participation is a product of financial literacy (Cardak and Wilkins 2009; Van Rooij et al. 2011), although it may also be affected by other factors (Yogo 2016). Financially literate individuals are more likely to participate in the financial market and invest in stocks (Almenberg and Dreber 2015; Van Rooij et al. 2011). Furthermore, Fong et al. (2020) proved that a one-unit increase in literacy score was associated with an 8.3% higher likelihood of stock market participation among adults aged 50 to 70 years in Singapore. Financially literate individuals are more likely to be savvy when it comes to managing their financial portfolios and dealing with risks (Hastings and Mitchell 2020).

Financial market participation enables a wide range of improvements in an individual's income and wealth positions (Calvet et al. 2007; Cocco and Gomes 2012), including saving for future retirement (Gustman and Steinmeier 2002; Farhi and Panageas 2007; Lim and Kwak 2016). Under a simple assumption, the fraction of stock market ownership is expected to be independent of age and wealth. However, using a more complicated model, Ameriks and Zeldes (2004) concluded that individuals do not decrease their equity holdings over time. As a matter of fact, a low stock-market-participation rate is persistent across different age cohorts. Moreover, people tend to participate in the financial market during a bullish market and are reluctant to do so during a bearish market (Mitchell and Utkus 2006). The New Zealand equity market is relatively small by international standards, with market capitalisation representing 30% of GDP, while the US and Australian markets accounted for 120% and 90% of their GDPs in 2014, respectively, according to the Reserve Bank of New Zealand.[17] On the other hand, the New Zealand equity market has performed well compared with other major equity markets, mainly due to the increase in liquidity provided by new market participants and the lower interest rate over the last decade. Many have believed that financial market participation enables individuals to accumulate sufficient wealth for their retirement; therefore, citizens in any nation should always be encouraged to participate in financial markets. Instead of indirect participation, direct financial market participation in our paper refers to an individual's own willingness to invest and control their investment in the financial market. We do not consider the voluntary pension scheme or the holding of bonds,[18] as they represent rather passive strategies. Direct financial market participation can capture an investor's risk preference and desire to invest, which may contribute towards retirement income. Therefore, we choose direct financial market participation as our main independent variable in our paper, and we conjecture that direct financial market participation can enhance retirement income in New Zealand.

## 3. Data and Methodology

### 3.1. New Zealand Household Economic Survey Data

Our dataset is collected from the New Zealand Household Economic Survey (HES) 2018 data. HES data are drawn from a cross-sectional survey conducted by Statistics NZ at the national level. The survey randomly selects households and individuals above 15 years old and collects information on household income, savings, expenditure and demographics, at both household and individual levels. HES survey questionnaires are designed to collect important household economic information and are frequently used by the central bank, the government and its agencies, and research institutions. For example, HES Expenditure data are used to estimate the New Zealand inflation rate calculated from the consumer price index (CPI). HES Income data are used to project the number of low-income families, and HES Savings data are used to calculate the net wealth of individuals/households. Data on household income and housing expenditure are updated on an annual basis, while data

on wealth and other expenditure are updated every 3 years. We only collected HES 2018 survey data for individual/household members who were 65 years old and above. Our sample contains 2,175 retiree observations, of which 1,050 households have one individual retiree, 552 households have two individual retirees, while the rest have more than two retirees within the same household.[19]

*3.2. Methodology*

The following regression is applied to examine the impact of direct financial market participation on retirement income:

$$RETIREMENT\_INCOME_i = \alpha_i + \beta_1 FINANCIAL\_MARKET\_PARTICIPATION\_IV_i + \beta_{i1} \sum_{i=1}^{n} Xi + \varepsilon_i \quad (1)$$

In Equation (1), $RETIREMENT\_INCOME_i$ is the dependent variable, measured by annuitised net wealth both with and without government pension as our objective measures, with overall life satisfaction and financial-situation satisfaction as our subjective measures. Our main independent variable, $FINANCIAL\_MARKET\_PARTICIPATION_i$, is a dummy variable equalling one if a retiree directly participates in the financial market and zero otherwise.[20] $X_i$ is a vector of control variables, including age, gender, living area, employment status, self-rated health, level of education, partnership status, risk aversion and gross retirement income.

On the one hand, a retiree who participates in the financial market can obtain a higher retirement income, while, on the other hand, retirees with better retirement incomes might have higher chances of participating in the financial market. In order to address this possible endogeneity issue, we follow Zhang et al. (2018) and randomly select an individual's financial-market-participation status in the same region to represent their direct financial-market-participation status, as retirees living in the same region are exposed to the same macroeconomic conditions and peer effects (Brown et al. 2008).[21] For example, if an observation retiree lives in the Auckland region according to his questionnaire, we randomly pick another retiree among the retiree samples living in the same region to represent his participation situation. Moreover, this randomly selected retiree's financial market participation is unlikely to correlate with the retiree's wealth accumulation, as there is little chance that these two retirees come from the same household, which largely eliminates endogeneity and omitted-variable issues. *Financial Market Participation_IV* in Equation (1) represents the randomly selected retiree's financial-market-participation situation in the same region and is considered as an instrumental variable (IV).

We select the responses from the survey questionnaires and define the relevant variables used in this paper, including individual net wealth, direct financial market participation, life satisfaction and financial-situation satisfaction. We use annuitised net wealth as our objective measure of retirement income (our dependent variable), as documented in Haveman et al. (2007a). Specifically, we first obtain the individual's estimated life expectancy based on New Zealand life tables and the individual's net wealth. Individual total net wealth consists of net wealth at the time of the survey and the present value of future government pensions. We calculate net wealth at the time of survey, which is the difference between assets and debts. The present value of a future government pension equals the total NZ Super for the remaining years of life. We use the current NZ Super as a proxy for future pension (annually standard-inflation-adjusted increases in pension offset the need to discount) and multiply the current pension by the number of remaining years of life to obtain the present value of the total future government pension. This overall individual net wealth is annuitised based on the inflation-adjusted Treasury-bill rate, considering the retiree's remaining life expectancy. This is the calculation for annuitised net wealth with a government pension. For annuitised net wealth without a government pension, it is calculated using the individual retiree's net wealth at the time of survey, discounted by the inflation-adjusted Treasury-bill rate, considering the retiree's remaining life expectancy. Using annuitised net wealth without a government pension enables us to examine the

significance of private savings with respect to retirement income. We also use the overall life satisfaction and financial-situation satisfaction of retirees as subjective measures to proxy for retirement income, similar to Bonsang and Klein (2012). Life satisfaction is a question asked of retirees in considering their overall life situation. It is assessed using a subjective rating score ranging from 1 to 5 (very dissatisfied to very satisfied). The second subjective measure is financial-situation satisfaction, which is a question about how retirees' incomes and needs are met, with a rating score ranging from 1 to 4 (not enough to more than enough).[22] Our independent variable is direct financial market participation, which is a dummy variable which equals one if a retiree receives income from equity holdings (dividends)[23], unit trusts and/or managed funds and is zero otherwise, following Brown et al. (2008).[24]

We also identify the following control variables: age, gender (male = 1, female = 0), living area (major urban area = 1, otherwise = 0), employment status (not working = 1, less than 30 h per week = 2, more than 30 h per week = 3), self-rated health (poor = 1, fair = 2, good= 3, very good = 4, excellent = 5), level of education (secondary school or below = 1, certificate or diploma = 2, university degree or above = 3), partnership status (with partner = 1, otherwise = 0), risk aversion (with home contents insurance = 1, otherwise = 0)[25], annual gross retirement income and self-controlled spending. A detailed description of all variables is presented in Table A1 in Appendix A.

Furthermore, our dependent variables are in continuous format for annuitised net wealth and in ordered format for overall life and financial satisfaction, while our instrumental independent variable takes binary values. Therefore, we apply non-linear regression in the first stage and linear regression in the second stage when the instrumental variable is used in the regression. The results may not be consistent due to the combined non-linear and linear regressions in both stages (Wooldridge 2002). Therefore, we use maximum-likelihood methods for both stages to ensure consistent and unbiased results. We also use extended regression models (ERMs) to address this issue, as documented in Cameron and Trivedi (2005). In addition, standard OLS, regional fixed effects, 2SLS and a three-stage model proposed by Adams et al. (2009) are also applied to conduct robustness tests.

## 4. Empirical Results

### 4.1. Summary Statistics

Table 1 presents the summary statistics of all variables for our sample observations used in this paper. The average annuitised net wealth with government pensions is NZD 44,802 and the annuitised net wealth without government pensions is NZD 17,854.[26] We find that 13% of our sample directly participate in the financial market. The average overall life satisfaction is 4.23 out of 5, and financial-situation satisfaction is 2.89 out of 4, indicating that overall life satisfaction is relatively higher than financial-situation satisfaction among New Zealand retirees. The average age of our sample retirees is 74 years old, and 49% are males. Based on answers to questions about the accessibility of various services, 67% of our sample retirees live in major urban cities, and the self-rated health condition is 3.42 out of 5. The employment status is 1.34 out of 3 (from zero as no employment to three as full-time employment). Further investigation of our data reveals that more than 21%[27] of retirees are still working either full-time or part-time. Regarding level of education, 19.1% of our sample retirees have a university degree or above, 36.2% have certificates or diplomas, and 44.7% have secondary school level education or below. We also observe that 62% of our sample retirees live with a partner and that 85% purchase home contents insurance, which is used as a proxy for risk aversion in the regression analysis. The average regular annual gross retirement income is approximately NZD 39,860.

**Table 1.** Summary Statistics.

|  | N | Mean | Std. Dev. | P1 | P99 |
|---|---|---|---|---|---|
| Overall_Annuitised_Net_Wealth | 2163 | 10.71 | 0.80 | 7.51 | 12.95 |
| Annuitised_Net_Wealth (without Government Pensions) | 2136 | 9.79 | 1.66 | 3.49 | 12.90 |
| Financial_Market_ Participation | 2175 | 0.13 | 0.34 | 0 | 1 |
| Overall_Life_Satisfaction (Measure 1) | 1848 | 4.23 | 0.79 | 1 | 5 |
| Financial_Situation_Satisfaction (Measure 2) | 1848 | 2.89 | 0.85 | 1 | 4 |
| Age | 2175 | 74.00 | 6.84 | 65 | 91 |
| Gender | 2175 | 0.49 | 0.50 | 0 | 1 |
| Living_Area | 2175 | 0.67 | 0.47 | 0 | 1 |
| Employment_Status | 2166 | 1.34 | 0.69 | 1 | 3 |
| Self-Rated_Health | 2148 | 3.42 | 1.08 | 1 | 5 |
| Highest_Education | 2133 | 1.75 | 0.76 | 1 | 3 |
| Partnership_Status | 2175 | 0.62 | 0.49 | 0 | 1 |
| Risk_Aversion | 1848 | 0.85 | 0.35 | 0 | 1 |
| Regular_Gross_Retirement _Income (in 1000) | 2175 | 39.86 | 55.09 | 0 | 227.41 |
| Spending_Range | 1815 | 4.25 | 1.10 | 1 | 5 |

This table presents the summary statistics of the 2175 ageing samples aged 65 years or above. Overall An-nuitised_Net_Wealth and Annuitised_Net_Wealth (without government pensions) are the natural logarithm values.[28]

*4.2. Regression Analysis: Objective Measurements*

4.2.1. Baseline Results: Annuitised Net Wealth

Table 2 presents our baseline results using annuitised net wealth as the dependent variable to measure retirement income.[29] We report results from annuitised net wealth both with and without government pensions. The coefficients on the financial market participation variable are positive and statistically significant at the 1% level for both models, revealing that retirees who directly participate in the financial market have 77.54%[30] more annuitised net wealth compared to those who do not directly participate.[31] This rate increases to 153.96%[32] when the government pension is not included in annuitised net wealth, indicating that direct financial market participation increases private savings, which contributes significantly to post-retirement income. Both results highlight the importance and significance of direct financial market participation with respect to the enhancement of retirement income. Following the literature (Hastings and Mitchell 2020), our results indicate an imperative obligation on the part of the New Zealand government to increase the likelihood of direct financial market participation via the promotion of financial literacy.

For the control variables in the first column (overall annuitised net wealth), the coeffi-cients on age, highest education, regular gross retirement income and risk aversion are all positive and statistically significant at the 1% level. We therefore conclude that older retirees with better education, higher regular gross retirement income and home contents insurance likely enjoy sufficient retirement income. On the other hand, we find that working and liv-ing with a partner reduces annuitised net wealth, as the coefficients on the employment and partnership status are negative and statistically significant at the 10% level. The coefficients on gender, self-rated health conditions and living area are all statistically insignificant, suggesting that these factors do not contribute to retirement income. The overall results for the second column (annuitised net wealth without government pensions) are similar, except we find that self-rated health condition and living with a partner have a positive impact on annuitised net wealth, as the coefficients are both positive and statistically significant at the 5% level. This highlights the importance of private savings for retirees with better self-rated health situations when the government pension is excluded. Retirees' living with

a partner is also positively related to retirement income. This finding is in stark contrast to the results obtained when overall annuitised net wealth is used to measure retirement income in Column 1, due to the unique setting of the New Zealand government pension system.[33]

**Table 2.** Baseline Objective Income Results.

|  | Overall Annuitised Net Wealth | Annuitised Net Wealth (without Government Pensions) |
|---|---|---|
| Financial_Market_Participation_IV | 0.574 *** | 0.932 *** |
|  | (13.58) | (11.63) |
| Age | 0.036 *** | 0.063 *** |
|  | (13.62) | (12.13) |
| Gender | 0.040 | −0.001 |
|  | (1.34) | (−0.02) |
| Living_Area | 0.033 | −0.030 |
|  | (1.12) | (−0.50) |
| Employment_Status | −0.074 * | −0.029 |
|  | (−2.33) | (−0.53) |
| Self-Rated_Health | 0.026 | 0.099 ** |
|  | (1.78) | (3.26) |
| Highest_Education | 0.078 *** | 0.241 *** |
|  | (3.71) | (5.92) |
| Partnership_Status | −0.074 * | 0.203 ** |
|  | (−2.35) | (3.12) |
| Regular_Gross_Retirement_Income | 0.005 *** | 0.007 *** |
|  | (6.81) | (6.58) |
| Risk_Aversion | 0.511 *** | 1.741 *** |
|  | (9.49) | (13.16) |
| Constant | 7.259 *** | 2.500 *** |
|  | (33.48) | (5.52) |
| Observations | 1821 | 1806 |

This table presents the regression results for post-retirement income measured by retirees' financial market participation, using an instrumental variable. Overall_Annuitised_Net_Wealth and Annuitised_Net_Wealth (without government pensions) are presented as logarithm values. The variable Financial_Market_Participation is a dummy variable that equals one if a retiree receives income or dividends from New Zealand company shares, managed funds and unit trusts and is zero otherwise. The control variables include: age, which represents the age of the retiree; gender, a dummy variable which equals one if the retiree is male and is zero otherwise; Living_Area, a dummy variable that equals one if the retiree lives in a major urban area and is zero otherwise; Employment_Status, which is equal to one if the retiree does not work, two if the retiree works less than 30 h a week and three if the retiree works more than 30 h a week; Self-Rated_Health, which is the retiree's self-perception of health condition, ranging from one to five, from poor to excellent; Highest_Educaiton, which is the highest education level achieved by the retiree at the time of the interview, which equals one if it is secondary school or below, two if the retiree has a certificate or diploma, and three if the retiree has a university degree; Partnership_Status, which is a dummy variable that equals one if the retiree has a partner and is zero otherwise; Regular_Gross_Retirement_Income, which is the retiree's regular annual retirement income before tax, in thousands of dollars; and Risk_Aversion, which is the individual's ownership of content insurance, which equals one if the retiree has home contents insurance and is zero otherwise. The *t*-statistics are in parentheses. ***, ** and * indicate statistical significance at the 1%, 5% and 10% levels, respectively.

### 4.2.2. Subsample Tests: Objective Measurements

We perform subsample tests in this section. Our sample is divided into subsample by age, gender, partnership status, living area and employment status. The results are reported in Table 3, where retirement income is measured by overall annuitised net wealth, and in Table 4, where retirement income is measured by annuitised net wealth without government pensions. Panel A shows the results for the age subsample, Panel B those for the gender subsample, Panel C those for the partnership-status subsample, Panel D those for the living-area subsample and Panel E those for the employment-status subsample.

**Table 3.** Subsample Tests for Overall Annuitised Net Wealth.

| | Overall Annuitised Net Wealth | | |
|---|---|---|---|
| **Subsample** | **Variable** | **Estimates** | **Observations** |
| Panel A: Age groups | | | |
| Younger retirees | Financial_Market_ Participation_IV | 0.753 *** (10.93) | 1056 |
| Older retirees | Financial_Market_ Participation_IV | 0.485 *** (5.96) | 1062 |
| Panel B: Gender groups | | | |
| Male | Financial_Market_ Participation_IV | 0.684 *** (10.66) | 1041 |
| Female | Financial_Market_ Participation_IV | 0.493 *** (7.03) | 1077 |
| Panel C: Partnership-status groups | | | |
| With partner | Financial_Market_ Participation_IV | 0.707 *** (11.55) | 1308 |
| Without partner | Financial_Market_ Participation_IV | 0.478 *** (3.94) | 810 |
| Panel D: Living-area groups | | | |
| Living in a major urban city | Financial_Market_ Participation_IV | 0.659 *** (9.92) | 1422 |
| Living in other cities | Financial_Market_ Participation_IV | 0.668 *** (7.74) | 693 |
| Panel E: Employment-status groups | | | |
| Not working | Financial_Market_ Participation_IV | 0.556 *** (9.84) | 1662 |
| Working | Financial_Market_ Participation_IV | 0.674 *** (6.12) | 192 |

This table presents the regression results for age, gender, partnership-status, living-area and employment-status subsamples. We only present results in this table for the Financial_Market_Participation_IV variable. We include all the control variables (age, gender, living area, employment status, self-rated health, level_of _education, partnership status, risk aversion and income) but do not tabulate them. Panel A reports the age groups for younger retirees (65–72 years old) and older retirees (73 years and older). Panel B reports the male and female subsample test results. Panel C reports the results for the subsamples of retirees living with a partner and without, respectively. Panel D reports the test results for the subsamples of retirees living in urban major and other cities. Panel E reports the test results for the subsamples of retiree employment status, where working includes both part-time and full-time working status. The *t*-statistics are in parentheses and *** indicates statistical significance at the 1%.

Panel A in Tables 3 and 4 presents the results for our age subsample. Our retirees are divided by the median age of 72 years old into a younger group and an older group. In Table 3, the coefficients on financial market participation are positive and statistically significant at the 1% level, with approximately 13% direct financial market participation for both age groups. Similar results are reported in Panel A, Table 4, with the coefficients on financial market participation also positive for both age groups. However, the coefficient for the younger age group is twice as large as that for the older age group, and it is statistically significant at the 1% level for both age groups. This indicates that direct financial market participation contributes a significant amount to retirement income for the younger retiree group compared to the older retiree group. Further investigation on risk aversion reveals that older retirees are more likely to purchase home contents insurance than younger retirees, indicating that the younger age group is less risk-averse.[34]

**Table 4.** Subsample Tests for Annuitised Net Wealth without Government Pensions.

| Subsample | Variable | Estimates | Observations |
|---|---|---|---|
| **Annuitized Net Wealth without Pension** | | | |
| *Panel A: Age groups* | | | |
| Younger retirees | Financial_Market_ Participation_IV | 1.438 *** (10.77) | 1035 |
| Older retirees | Financial_Market_ Participation_IV | 0.751 *** (4.97) | 1056 |
| *Panel B: Gender groups* | | | |
| Male | Financial_Market_ Participation_IV | 1.146 *** (9.77) | 1026 |
| Female | Financial_Market_ Participation_IV | 0.918 *** (5.70) | 1065 |
| *Panel C: Partnership-status groups* | | | |
| With partner | Financial_Market_ Participation_IV | 1.178 *** (12.21) | 1293 |
| Without partner | Financial_Market_ Participation_IV | 0.872 *** (4.25) | 798 |
| *Panel D: Living-area groups* | | | |
| Living in a major urban city | Financial_Market_ Participation_IV | 1.109 *** (8.29) | 1407 |
| Living in other cities | Financial_Market_ Participation_IV | 1.134 *** (8.16) | 684 |
| *Panel E: Employment-status groups* | | | |
| Not working | Financial_Market_ Participation_IV | 0.957 *** (7.97) | 1641 |
| Working | Financial_Market_ Participation_IV | 1.247 ** (6.45) | 189 |

This table presents the regression results for age, gender, partnership-status, living-area and employment-status subsamples. We only present results in this table for the Financial_Market_Participation_IV variable. We include all the control variables (age, gender, living area, employment status, self-rated health, level_of _education, partnership status, risk aversion and income) but do not tabulate them. Panel A reports the age groups for younger retirees (65–72 years old) and older retirees (73 years and older). Panel B reports the male and female subsample test results. Panel C reports the results for the subsamples of retirees living with a partner and without, respectively. Panel D reports the test results for the subsamples of retirees living in urban major and other cities. Panel E reports the test results for the subsamples of retiree employment status, where working includes both part-time and full-time working status. The *t*-statistics are in parentheses. *** and ** indicate statistical significance at the 1% and 5% levels, respectively.

Panel B presents the results for the gender subsample. Säve-Söderbergh (2012) documents that males are more risk-taking than females and have a higher financial-market-participation rate of 14.8%. Female retirees, on the other hand, have an 11.1% participation rate. Our results are consistent with the findings documented in Säve-Söderbergh (2012), with the coefficients on financial market participation positive and statistically significant at the 1% level for the male group, with greater influence from financial market participation, indicating that male retirees are more likely to participate in the financial market, resulting in a greater retirement income. This is consistent with the result for the overall sample that 60% of our retiree financial participants are male.

Panel C in both tables presents the results from the partnership-status subsample. We conclude that retirees living with a partner or not so living benefit from financial market participation, as the coefficients are positive and statistically significant at the 1% level. The results in Panel D for the living-area subsample suggest that financial market participation contributes to a significant increase in retirement income, regardless of living area. Similar results are found for the employment-status subsample in Panel E, such that, regardless of

employment status, financial participation contributes significantly to an improvement in retirement income.

### 4.2.3. Robustness Tests: Alternative Approaches

In this section, we use several alternative approaches to conduct robustness checks, including standard OLS, standard OLS adjusted for region fixed effects, 2SLS and three-stage regression (Adams et al. 2009). The OLS model illustrates a fundamental linear relationship between the variables, with standard OLS adjusted for region fixed effects also used to eliminate the variations among different regions, while 2SLS is used to deal with any possible endogeneity issues. In addition, we follow Adams et al. (2009) and use three-stage OLS to deal with the binary endogenous independent and instrumental variables. The results of these alternative models are reported in Table 5. All of the coefficients on financial market participation are positive and statistically significant at the 1% level, confirming our baseline results in Table 2 showing that financial market participation enhances retirement income. This phenomenon is stronger when we only consider the pre-retirement income of retirees, as the magnitude of the coefficients on annuitised net wealth without government pensions is greater than those on overall annuitised net wealth, across all alternative models. Therefore, the results in Table 5 are consistent with our baseline results and confirm that our main results are robust. Financial market participation enhances retirees' overall annuitised net wealth, and this impact is strengthened when the government pension is excluded when calculating annuitised net wealth.

**Table 5.** Alternative Methods for Objective Measurements.

| | OLS | | Region Fixed Effects | | 2SLS | | Three-Stage Strategy | |
|---|---|---|---|---|---|---|---|---|
| | Overall ANW | ANW | Overall ANW | ANW | Overall ANW | ANW | Overall ANW | ANW |
| Financial_ Market_ Participation (_IV) | 0.404 *** | 0.637 *** | 0.404 *** | 0.639 *** | 0.550 *** | 0.852 *** | 0.544 *** | 0.797 *** |
| | (9.36) | (8.96) | (9.15) | (7.20) | (9.62) | (7.47) | (9.71) | (7.11) |
| Age | 0.036 *** | 0.063 *** | 0.035 *** | 0.062 *** | 0.036 *** | 0.063 *** | 0.036 *** | 0.063 *** |
| | (13.72) | (12.22) | (15.48) | (13.41) | (15.50) | (13.47) | (15.51) | (13.50) |
| Gender | 0.041 | 0.001 | 0.038 | −0.006 | 0.040 | −0.001 | 0.040 | −0.000 |
| | (1.38) | (0.01) | (1.29) | (−0.09) | (1.35) | (−0.01) | (1.35) | (−0.01) |
| Living_Area | 0.035 | −0.026 | 0.001 | −0.093 | 0.033 | −0.029 | 0.033 | −0.028 |
| | (1.18) | (−0.45) | (0.03) | (−1.47) | (1.09) | (−0.47) | (1.09) | (−0.46) |
| Employment_ Status | −0.086 ** | −0.050 | −0.081 ** | −0.037 | −0.076 ** | −0.035 | −0.077 ** | −0.039 |
| | (−2.69) | (−0.91) | (−3.16) | (−0.73) | (−2.95) | (−0.67) | (−2.97) | (−0.75) |
| Self-Rated_Health | 0.029 * | 0.104 *** | 0.027 | 0.100 *** | 0.027 | 0.100 *** | 0.027 | 0.101 *** |
| | (1.98) | (3.44) | (1.94) | (3.55) | (1.87) | (3.50) | (1.88) | (3.54) |
| Highest_ Education | 0.088 *** | 0.259 *** | 0.085 *** | 0.252 *** | 0.079 *** | 0.246 *** | 0.079 *** | 0.249 *** |
| | (4.21) | (6.41) | (4.30) | (6.31) | (3.94) | (6.06) | (3.96) | (6.15) |
| Partnership_ Status | −0.066 * | 0.217 *** | −0.075 * | 0.202 ** | −0.073 * | 0.207 *** | −0.072 * | 0.209 *** |
| | (−2.10) | (3.32) | (−2.43) | (3.25) | (−2.34) | (3.32) | (−2.34) | (3.36) |
| Risk_ Aversion | 0.521 *** | 1.759 *** | 0.540 *** | 1.799 *** | 0.512 *** | 1.746 *** | 0.513 *** | 1.749 |
| | (9.59) | (13.22) | (12.47) | (20.31) | (11.87) | (19.79) | (11.88) | (19.85) |
| Regular_ Gross_ Retirement_ Income | 0.006 *** | 0.008 *** | 0.006 *** | 0.008 *** | 0.005 *** | 0.008 *** | 0.005 *** | 0.008 *** |
| | (7.08) | (6.99) | (14.88) | (10.38) | (13.66) | (9.53) | (13.72) | (9.71) |
| Constant | 7.225 *** | 2.441 *** | 7.286 *** | 2.546 *** | 7.255 *** | 2.484 *** | 7.253 *** | 2.473 *** |
| | (33.35) | (5.40) | (37.25) | (6.42) | (36.75) | (6.22) | (36.75) | (6.19) |
| R-squared | 0.37 | 0.40 | | | | | | |
| Observations | 1821 | 1806 | 1821 | 1806 | 1821 | 1806 | 1821 | 1806 |

This table presents four alternative regression methods for objective income measurements. The first one is ordinary least squares, the second one is region fixed effects, the third is 2SLS and the last one is a three-stage strategy. All of these methods have been applied for overall annuitised net wealth (overall ANW) and annuitised net wealth without government pensions (ANW). The *t*-statistics are in parentheses. ***, ** and * indicate statistical significance at the 1%, 5% and 10% levels, respectively.

### 4.3. Regression Analysis: Subjective Measurements

4.3.1. Baseline Results

Two of the survey questions are designed to rate overall life satisfaction and financial-situation satisfaction. Therefore, we use these as our subjective measures to proxy for retirement income in our regression, with the results reported in Table 6. In Table 6, the first model is the regression for overall life satisfaction and the second model is the regression for financial-situation satisfaction. We find that the coefficient on financial market participation is positive and statistically significant at the 1% level in the second model, while it is statistically insignificant in the first model. This indicates that direct financial market participation enhances financial well-being. The results for all control variables are largely in line with our main results when using annuitised net wealth as a proxy for retirement income. Therefore, using subjective measurements, our results are consistent with those obtained when using objective measurements.

**Table 6.** Baseline Subjective Income Results.

|  | Overall Life Satisfaction | Financial-Situation Satisfaction |
|---|---|---|
| Financial_Market_ Participation_IV | 0.036 | 0.497 *** |
|  | (0.33) | (4.48) |
| Age | 0.009 * | 0.022 *** |
|  | (2.17) | (5.52) |
| Gender | −0.083 | −0.083 |
|  | (−1.51) | (−1.56) |
| Living_Area | −0.038 | −0.025 |
|  | (−0.68) | (−0.45) |
| Employment_Status | 0.086 | 0.077 |
|  | (1.66) | (1.20) |
| Self-Rated_Health | 0.295 *** | 0.175 *** |
|  | (10.55) | (6.67) |
| Highest_Education | 0.028 | 0.131 *** |
|  | (0.76) | (3.50) |
| Partnership_Status | 0.319 *** | 0.248 *** |
|  | (5.66) | (4.21) |
| Regular_Gross_ Retirement_Income | 0.000 | 0.009 *** |
|  | (0.38) | (3.61) |
| Risk_Aversion | 0.103 | 0.310 *** |
|  | (1.36) | (3.86) |
| Observations | 1827 | 1827 |

This table presents the regression results for subjective income measurements. One is overall life satisfaction as regards life, and the other is financial-situation satisfaction considering financial incomes and needs. All the control variables are the same as in the objective measurement results. The *t*-statistics are in parentheses. *** and * indicate statistical significance at the 1% and 10% levels, respectively.

4.3.2. Subsample Tests: Subjective Measurements

In this section, we perform subsample tests using the subjective measures to proxy for retirement income. Table 7 presents our results: Panel A for the age group, Panel B for the gender group, Panel C for the partnership-status group, Panel D for the living-area group and Panel E for the employment-status group. We only conduct analyses using financial-situation satisfaction, as the impact of financial participation on overall life satisfaction is insignificant, as reported in Table 6. In addition, financial-situation satisfaction is a better subjective proxy for measuring an individual's financial well-being.[35] Our results in Table 7 are basically consistent with the results reported in Table 6. In addition, we find that younger retirees, male retirees, retirees without a partner, retirees living in major cities and retirees not currently working have significantly higher financial-situation satisfaction due to actively participating in the financial market. Our results in Tables 3 and 4 document that retirees with a partner, living in other cities and currently working have greater annuitised net wealth with active financial market participation, while retirees without a partner,

living in major urban cities and currently not working enjoy greater financial-situation satisfaction, as shown in Table 7. Moreover, male younger retirees with financial market participation have greater annuitised wealth and higher financial-situation satisfaction.

**Table 7.** Subsample Tests for Subjective Financial-Situation Satisfaction.

| Subsample | Variable | Estimates | Observations |
|---|---|---|---|
| **Financial-Situation Satisfaction** | | | |
| Panel A: Age groups | | | |
| Younger retirees | Financial_Market_ Participation_IV | 0.368 *** (3.34) | 870 |
| Older retirees | Financial_Market_ Participation_IV | 0.348 *** (3.83) | 960 |
| Panel B: Gender groups | | | |
| Male | Financial_Market_ Participation_IV | 0.362 *** (4.20) | 861 |
| Female | Financial_Market_ Participation_IV | 0.305 ** (2.90) | 969 |
| Panel C: Living-status groups | | | |
| With partner | Financial_Market_ Participation_IV | 0.342 *** (4.19) | 1110 |
| Without partner | Financial_Market_ Participation_IV | 0.444 *** (3.74) | 720 |
| Panel D: Living-area groups | | | |
| Living in a major urban city | Financial_Market_ Participation_IV | 0.398 *** (4.97) | 1206 |
| Living in other cities | Financial_Market_ Participation_IV | 0.307 * (2.49) | 624 |
| Panel E: Employment-status groups | | | |
| Not working | Financial_Market_ Participation_IV | 0.434 *** (5.60) | 1470 |
| Working | Financial_Market_ Participation_IV | 0.100 (0.37) | 162 |

This table presents the regression results for age, gender, living-status, living-area and employment-status subsamples. We only present results in this table for the Financial_Market_Participation_IV variable. We include all the control variables (age, gender, living area, employment status, self-rated health, highest education, living status, risk aversion and income) but do not tabulate them. Panel A reports the age groups for younger retirees (65–72 years old) and older retirees (73 years and older). Panel B reports the male and female subsample test results. Panel C reports the results for the subsamples of retirees living with a partner and without, respectively. Panel D reports the test results for the subsamples of retirees living in urban major and other cities. Panel E reports the test results for the subsamples of retiree employment status, where working includes both part-time and full-time working status. The *t*-statistics are in parentheses. ***, ** and * indicate statistical significance at the 1%, 5% and 10% levels, respectively.

### 4.3.3. Robustness Tests: Subjective Measurements

We apply alternative models and survey questions to conduct further analysis for the robustness checks in this section. Firstly, the ordered probit model is used as an alternative model to examine the impact of financial market participation on overall life satisfaction and financial-situation satisfaction. The results are reported in the first two columns in Table 8. The results for the alternative model are consistent with our results reported in Table 6. The coefficient of financial market participation is positive and statistically significant at the 1% level, when financial-situation satisfaction is used to proxy for subjective retirement income. Secondly, we use another question in the survey to define an alternative variable, "self-controlled spending", to proxy for retirement income as the dependent variable. This survey question asks how much money, on average, a retiree has each week for spending on things without consulting anyone else.[36] The results are reported in Table 8, Column

3 for the ordered probit model and Column 4 for the extended regression model. The coefficients on financial market participation are positive and statistically significant at the 5% and 1% levels for both models. Overall, the results in Table 8 are consistent with our baseline results, regardless of the model and the proxy used. Therefore, we conclude that financial market participation can improve retirees' subjective financial well-being.

**Table 8.** Robustness Checks for Subjective Measurements.

| | Alternative Method | | Alternative Question | |
| --- | --- | --- | --- | --- |
| | Ordered Probit Model | | Ordered Probit Model | Extended Regression Model |
| | Overall Life Satisfaction | Financial-Situation Satisfaction | Self-Controlled Spending | Self-Controlled Spending |
| Financial_Market_Participation(_IV) | 0.057 | 0.410 *** | 0.430 ** | 0.552 *** |
| | (0.67) | (4.55) | (4.34) | (4.85) |
| Age | 0.009 * | 0.023 *** | 0.015 ** | 0.015 *** |
| | (2.16) | (5.54) | (3.38) | (3.37) |
| Gender | −0.083 | −0.082 | 0.045 | 0.045 |
| | (−1.51) | (−1.55) | (0.77) | (0.77) |
| Living_Area | −0.038 | −0.024 | −0.183 ** | −0.184 ** |
| | (−0.69) | (−0.44) | (−2.96) | (−2.97) |
| Employment_Status | 0.088 | 0.071 | 0.158 * | 0.166 * |
| | (1.70) | (1.11) | (2.36) | (2.46) |
| Self-Rated_Health | 0.295 *** | 0.176 *** | 0.102 ** | 0.101 *** |
| | (10.54) | (6.73) | (3.54) | (3.51) |
| Highest_Education | 0.027 | 0.136 *** | 0.096 * | 0.089 * |
| | (0.73) | (3.65) | (2.35) | (2.20) |
| With_Partner | 0.318 *** | 0.252 *** | −0.112 | −0.118 |
| | (5.64) | (4.28) | (−1.83) | (−1.93) |
| Regular_Gross_Retirement_Income | 0.000 | 0.009 *** | 0.003 | 0.003 |
| | (0.35) | (3.71) | (1.46) | (1.33) |
| Risk_Aversion | 0.102 | 0.315 *** | 0.208 ** | 0.201 * |
| | (1.35) | (3.92) | (2.58) | (2.49) |
| Observations | 1827 | 1827 | 1791 | 1791 |

This table presents an alternative model and an alternative survey question for subjective retirement income measurements. An ordered probit model is used as the alternative method, and the personal dispensable spending range is the alternative survey question used for subjective income measurement. The *t*-statistics are in parentheses. ***, ** and * indicate statistical significance at the 1%, 5% and 10% levels, respectively.

## 5. Conclusions

Using a manually collected dataset from the 2018 New Zealand Household Economic Survey, this paper has examined the impact of direct financial market participation on retirement income. We used annuitised net wealth with and without government pensions as our objective measures to proxy for retirement income. We also used retirees' overall life satisfaction and financial-situation satisfaction as subjective measures to proxy for retirement income. From a sample of 2175 retirees, we conclude that direct financial market participation has a positive and significant impact on retirement income when using both objective and subjective measures. We found that retirees who directly participate in the financial market experience an approximately 78% increase in annuitised wealth compared to those who do not. This rate increases substantially to 154% when the government pension is not included in the calculation of annuitised net wealth, indicating that direct financial market participation increases private savings, which contributes significantly to post-retirement income. Our research confirms that financial market participation improves individual wealth and income (Calvet et al. 2007; Cocco and Gomes 2012) among retirees. Further investigation from the baseline results for control variables revealed that older

retirees with higher regular gross retirement income, better education and home contents insurance are likely to have greater retirement incomes in terms of both objective measures. On the one hand, we found that living with a partner and working reduce retirement income as measured by annuitised net wealth with government pensions. On the other hand, retirees with a better self-rated health situation and living with a partner enjoy greater annuitised net wealth without government pensions. Other factors, such as gender and living area, have no impact on retirement income.

We further conducted our analysis using various subsamples, including age, gender, partnership-status, living-area and employment-status subsamples. Our results highlight the different influence channels between different partnership-status, living-area and employment-status groups; for example, the impact of financial market participation on retirement income is reflected in the objective annuitised wealth channel for retirees with a partner, living in other cities and currently working. On the other hand, it is reflected in the subjective financial well-being channel for retirees without a partner, living in major urban cities and currently not working. Other subsample results show that younger male retirees are more likely to participate in the financial market, resulting in a greater retirement income in terms of both objective and subjective measures. We used several alternative approaches to conduct robustness checks, and all results were found to be consistent with our baseline results, suggesting that our main results are robust.

This paper addresses an important issue relating to retirement income and direct financial market participation. Scobie et al. (2004) found that New Zealand superannuation provides a floor for those pre-retirees who are in the lowest 40% of the income distribution and reduces inequality in retirement wealth accumulation. According to the OECD, the net pension replacement rate in New Zealand was 43% of pre-retirement male earnings in 2020 compared with the OECD average of 63%.[37] Noviarini et al. (2021) showed that the New Zealand pension system puts great pressure on retirees to personally manage their wealth, while 70% of retirees are unprepared for retirement, with only 43% understanding the different strategies and approaches to investing their money. Our paper documents a timely and effective solution enabling retirees to enhance their post-retirement lifestyles. We find that New Zealand retirees benefit significantly from participating in the financial market post-retirement. However, a low financial-market-participation rate has been an issue worldwide (Grinblatt et al. 2011), and financial literacy is documented to be an important and effective factor in improving financial market participation (Cardak and Wilkins 2009; Van Rooij et al. 2011). Therefore, it is suggested that policy makers in New Zealand carry imperative obligations to promote financial literacy in order to increase the likelihood of direct financial market participation. Furthermore, our paper has important policy implications and sheds extra light on the significance of financial market participation in helping individuals obtain a better and more secure retirement life.

**Author Contributions:** Conceptualization, X.X. and J.F.; methodology, X.X. and J.F.; software, X.X.; validation, X.X. and M.Y.; formal analysis, X.X.; investigation, X.X.; resources, X.X.; data curation, X.X.; writing—original draft preparation, X.X.; writing—review and editing, L.Z.; visualization, L.Z.; supervision, M.Y.; project administration, X.X. All authors have read and agreed to the published version of the manuscript.

**Funding:** This research received no external funding.

**Informed Consent Statement:** Patient consent was waived because it is secondary data from Stats NZ and the authors did not carry out the surveys originally.

**Data Availability Statement:** Data is unavailable due to privacy restrictions and access might be granted once applying from Stats NZ.

**Conflicts of Interest:** The authors declare no conflict of interest.

## Appendix A

**Table A1.** New Zealand Household Economic Survey Variable Definitions.

| Variables | Description | Observations |
|---|---|---|
| Overall Annuitised Net Wealth | The present value of the remaining years of New Zealand government pensions (including NZ superannuation, veteran's pension, war disablement pension, surviving spouse pension and other types of New Zealand government pension) are added to individual net wealth in 2018. The final value is the individual's overall net wealth, and this value is annuitised based on remaining life expectancy (according to gender and region) and the discount rate (inflation-adjusted T-bill rate in 2018). The natural logarithm of overall annuitised net wealth is the final value for this variable. | 2163 |
| Annuitised Net Wealth (without Government Pensions) | This value is the logarithm of annuitised net wealth without government pensions. Annuitised net wealth is overall net wealth (at the interview time) annuitised based on the remaining life expectancy and the discount rate. | 2136 |
| Overall Life Satisfaction | Survey question: I'm now going to ask you a very general question about your life. This includes all areas of your life, not just what we have talked about so far. Very dissatisfied = 1, dissatisfied = 2, neither satisfied nor dissatisfied = 3, satisfied = 4, very satisfied = 5. For this question, only one person from each household answered the question, and we duplicated the answers for other members in the same household. | 1848 |
| Financial-Situation Satisfaction | Survey question: I would like you to think about how well (you and your partners combined) total income meets your everyday needs for such things as accommodation, food, clothing and other necessities. Not enough = 1, only just enough = 2, enough = 3, more than enough = 4. For this question, only one person from each household answered the question, and we duplicated the answers for other members in the same household as well. | 1848 |
| Financial Market Participation | Dummy variable, direct financial market participation = 1, otherwise = 0. Income from New Zealand company dividends, unit trusts and managed funds. | 2175 |
| Age | Individuals were 65 years or above in the 2017–2018 survey period. | 2175 |
| Gender | Dummy variable, male = 1, female = 0. | 2175 |
| Living Area | Living in major urban areas = 1, living elsewhere = 0, depending on the accessibility to services, according to the urban area classification in New Zealand. | 2175 |
| Employment Status | Not working = 1, working less than 30 h = 2, working more than 30 h = 3. | 2166 |
| Highest Education | The highest education level has been achieved by the 2017–2018 interview. Secondary school or below = 1, certificate or diploma = 2, university degree = 3 | 2133 |
| Self-Rated Health | Self-assessed health status. Poor = 1, fair = 2, good = 3, very good = 4, excellent = 5. | 2148 |
| Partnership Status | Dummy variable, have a partner = 1, otherwise = 0. | 2175 |
| Risk Aversion | Do you have home contents insurance? Yes = 1, no = 0. For this question, only one person from each household answered the question, and we duplicated the answers for other members in the same household as well. | 1848 |

**Table A1.** *Cont.*

| Variables | Description | Observations |
|---|---|---|
| Gross Retirement Income | Total annual personal income from regular and recurring sources, including income from current and previous employment, investment income (rental properties, interests, dividends), all types of government transfer and other regular and recurring income sources (private superannuation payment, spousal and child support payments, etc.) before taxes in the 2017–2018 survey. | 2175 |
| Spending Range | About how much money, on average, do you have each week for spending on things for yourself without consulting anyone else? None = 1, under \$10 = 2, \$10–\$25 = 3, \$26–\$50 = 4, more than \$50 = 5. For this question, only one answer was obtained from each household, and we duplicated the answers for other members in the same household as well. | 1815 |

This table presents the descriptions of each variable and related questions in the 2018 HES New Zealand survey. The sample comprises those aged 65 years and above, and there are 2175 observations overall.

**Table A2.** First-Stage Results for the Baseline Objective Retirement Income Measures.

| | Stage 1 Results for Overall Annuitised Net Wealth Financial_Market_Participation | Stage 1 Results for Annuitised Net Wealth (without Government Pensions) Financial_Market_Participation |
|---|---|---|
| Financial_Market_ Participation_IV | 2.847 *** (22.01) | 2.828 *** (21.91) |
| Age | −0.003 (−0.37) | −0.005 (−0.58) |
| Gender | 0.061 (0.51) | 0.073 (0.62) |
| Living_Area | 0.032 (0.26) | 0.025 (0.20) |
| Employment_ Status | −0.120 (−1.12) | −0.124 (−1.18) |
| Self-Rated_ Health | 0.128 * (2.16) | 0.127 * (2.15) |
| Highest_Education | 0.185 * (2.30) | 0.175 * (2.20) |
| Partnership_Status | 0.437 *** (3.58) | 0.397 *** (3.36) |
| Regular_Gross_ Retirement_ Income | 0.004 ** (3.09) | 0.004 ** (3.22) |
| Risk_Aversion | 0.768 ** (3.04) | 0.657 *** (2.70) |
| Constant | −3.599 *** (−4.44) | −3.264 *** (−3.99) |
| Observations | 1821 | 1806 |

This table reports the first-stage relation between Financial_Market_Participation and Financial_Market_Participation_IV, which is the randomly selected retiree's financial-market-participation situation for the baseline objective retirement income measures. All the control variables are the same as for the baseline results in Table 2. The *t*-statistics are in parentheses. ***, ** and * indicate statistical significance at the 1%, 5% and 10% levels, respectively.

**Table A3.** First-Stage Results for the Subjective Financial Well-Being Measures.

| | Stage 1 Results for Overall Life Satisfaction Financial_Market_Participation | Stage 1 Results for Financial-Situation Satisfaction Financial_Market_Participation |
|---|---|---|
| Financial_Market_ Participation_IV | 2.873 *** (21.97) | 2.875 *** (22.08) |
| Age | −0.000 (−0.04) | −0.001 (−0.15) |
| Gender | 0.087 (0.73) | 0.081 (0.68) |
| Living_Area | 0.024 (0.20) | 0.035 (0.29) |
| Employment_Status | −0.109 (−1.04) | −0.117 (−1.10) |
| Self-Rated_Health | 0.130 * (2.16) | 0.135 * (2.26) |
| Highest_Education | 0.173 * (2.16) | 0.169 * (2.11) |
| Partnership_Status | 0.394 *** (3.38) | 0.402 *** (3.44) |
| Regular_Gross_ Retirement_Income | 0.004 ** (3.22) | 0.004 ** (3.27) |
| Risk_Aversion | 0.901 ** (3.03) | 0.855 ** (2.92) |
| Constant | −3.873 *** (−4.65) | −3.769 *** (−4.52) |
| Observations | 1827 | 1827 |

This table reports the first-stage relation between Financial_Market_Participation and Financial_Market_Participation_IV, which is the randomly selected retiree's financial-market-participation situation for the subjective financial well-being measures. All the control variables are the same as for the baseline results in Table 3. The *t*-statistics are in parentheses. ***, ** and * indicate statistical significance at the 1%, 5% and 10% levels, respectively.

## Notes

[1] https://data.oecd.org/pop/elderly-population.htm (accessed on 1 December 2020).

[2] More details about the three-pillar pension system are discussed in Section 2.1.

[3] https://www.stuff.co.nz/business/119288609/pension-age-debate-crunch-coming (accessed on 10 March 2021).

[4] https://www.mercer.com/our-thinking/global-pension-index-2021.html?mkt_tok=NTIxLURFVi01MTMAAAGAd1aOK2i2tzei9r7fZTfu7s98i6YLVL8YytHDtsXCxuuzGmJJbWLDj4ygUAR2DVXlbxkv5MbgEvc953cdhSwotsHu8kXGIt7-3ppc197oxfenzfHHug (accessed on 20 Feburary 2022).

[5] https://www.beehive.govt.nz/release/nz-superannuation-age-lift-67-2040 (accessed on 1 December 2020).

[6] Twenty-six countries were studied by Giannetti and Koskinen (2010); New Zealand's participation rate reached 31%, only behind Australia (40.4%) and ahead of the UK (30%), Japan (29.7%), Denmark (28%) and the US (26%).

[7] https://www.fma.govt.nz/assets/Reports/Growing-New-Zealands-Capital-Markets-2029.pdf (accessed on 15 March 2021).

[8] https://www.treasury.govt.nz/sites/default/files/2017-03/sup-3753700.pdf (accessed on 10 December 2020).

[9] The New Zealand superannuation payment differs according to partnership status and a person's dependent-child situation. These payments are the highest levels for those living with a dependent child.

[10] https://cffc.govt.nz/news-and-media/news/purpose-of-nz-retirement-income-system-defined/ (accessed on on 25 Feburary 2022).

[11] https://www.massey.ac.nz/massey/fms/Colleges/College%20of%20Business/School%20of%20Economics%20&%20Finance/FinEd/documents/RetExpBudget_Report_June2021.pdf (accessed on 9 March 2022).

[12] According to World Bank categorisation, KiwiSaver belongs to the third pillar as it is voluntary. However, due to its auto-enrolment and opt-out feature, it could be considered as a combination of the second and third pillars. We treat it as the second pillar in NZ pension system to make the pillar levels consistent in this paper. Otherwise, the NZ pension system does not have the mandatory employee contribution second pillar.

[13] According to the Inland Revenue Department and Statistics NZ, by May 2015, the participation rate for those aged 18 to 64 years reached 76.5% and the opt-out rate was 9.3%.

14 https://www.fsc.org.nz/site/fsc1/Reports/Kiwisaver%20&%20the%20wealth%20of%20NZers-NZIER%20report%20to%20FSC,%20Aug15.pdf (accessed on 15 December 2020).

15 https://www.stats.govt.nz/reports/housing-in-aotearoa-2020 (accessed on 14 July 2022).

16 https://cffc-assets-prod.s3.ap-southeast-2.amazonaws.com/public/Uploads/Corporate-reports/Statement-of-Intent/CFFC-Statement-of-Intent-2021-2024.pdf (accessed on 14 July 2022).

17 https://www.rbnz.govt.nz/financial-stability/financial-stability-report/fsr2014-11/the-role-of-capital-markets-in-the-new-zealand-financial-system#fn3 (accessed on 1 December 2020).

18 Moreover, our results can be lifted by including these passive operations as well and, due to the availability of the data, the passive methods are not included.

19 Due to confidentiality issues, we could not give detailed compositions of the remaining households.

20 Cameron et al. (2007) showed that New Zealand firms have dividend payout ratios that are significantly higher than other countries. Therefore, dividend is a good proxy for financial participation among retirees.

21 The regions are Northland Region, Auckland Region, Waikato Region, Bay of Plenty Region, Gisborne Region, Hawke's Bay Region, Taranaki Region, Manawatu–Wanganui Region, Wellington Region, West Coast Region, Canterbury Region, Otago Region, Southland Region, Tasman Region, Nelson Region and Marlborough Region, according to the questionnaire region classification.

22 If only one member in a household answered these questions, the same score was assigned to other members within the same household.

23 The equity holdings are limited to New Zealand domestic stock, and overseas financial market participation is not included. On the one hand, the NZ equity market is less volatile, so the investment is more aligned with retirees' risk preferences. On the other hand, NZ retirees are more familiar with domestic markets and so prefer to invest domestically due to home bias. Moreover, QuayStreet Asset Management also reported the relative returns from key investment assets for 1998 to 2008, explaining that investing in local companies does not cut investors out of the global action; many NZ listed firms, such as Fisher & Paykel Healthcare, Fonterra and the A2 Milk Company, do much of their business offshore. Additionally, manged funds can capture part of overseas investment as well.

24 This variable is from three income sources, which are income from dividends from New Zealand companies, income from returns of unit trusts and income from returns of managed funds, according to the income classification in the questionnaire.

25 Home contents insurance may be confounded by homeownership. However, in our data, homeownership is 60%, while home contents insurance reaches 85%. This shows that retirees would buy home contents insurance regardless of their homeownership status.

26 The annuitised net wealth presented in Table 1 is in logarithm values. We therefore convert these values into $\exp^\wedge(10.71) = 44{,}802$ and $\exp^\wedge(9.79) = 17{,}854$, respectively.

27 This number is not reported in Table 1.

28 Due to confidentiality rules, we can only present 1% and 99% of the values instead of minimum and maximum values.

29 The first-stage results are presented in Appendix A, Table A2 and suggest that the instrumental variable is valid.

30 $\exp^\wedge(0.574) - 1 = 77.54\%$.

31 We also performed the same regressions without controlling individual net wealth to prevent endogeneity issues. The results are similar and are available upon request.

32 $\exp^\wedge(0.932) - 1 = 153.96\%$.

33 The pension paid to a couple is less than the sum of two individual payments.

34 The statistics on risk aversion are not presented in Tables 3 and 4, but are available upon request.

35 We also conducted subjective life-satisfaction subsample tests, the results being the same as the main results. No group had statistically significant impacts.

36 None = 1, under $10 = 2, $10–$25 = 3, $26–$50 = 4, more than $50 = 5.

37 https://data.oecd.org/pension/net-pension-replacement-rates.htm (accessed on 3 November 2021).

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
