# Peer review of "Retirement Income and Financial Market Participation in New Zealand†"

_ijfs, doi:10.3390/ijfs11010024_

Round 1

Reviewer 1 Report

The aim of this research is to investigate the influence of direct market participation of retirement outcomes for New Zealanders.  It uses a retired persons current participation in the stock market to indicate past participation in the stock market.  It them measures retirement outcomes by objective measures, annuitized net wealth, and subjective measures, self-reported financial satisfaction.  

The paper is very well written, and some thorough analysis has been conducted.  The authors have completed rigorous research and should be commended on their work.  However, I do have major concerns regarding the paper.  I aim to give the authors a chance to address these concerns either by changing the framing of the paper or by conducting different analysis.  

A major concern I have is the independent variable and dependent variables are inherently linked. Participation in the stock market is correlated with wealth and your dependent variables (annuitized net wealth and financial satisfaction) are also inherently related to wealth.  Your models also include net wealth as a control variable which has a large correlation with your dependent variables.  The relationship between net wealth and participation in the stock market is not reported in the paper (perhaps include a correlation table as an appendix). There are two main problems with this.  Firstly, the argument tends to descend to a chicken or the egg debate.  Am I wealthy because I participate in the stock market, or do I participate in the stock market because I am wealthy?  This paper does not appear to unpack this argument at all.  Secondly, an implication of your results is that more people should participate in the stock market when planning for retirement.  This argument is not clear cut because you cannot show causation.  Yet, when the paper uses terms like ‘impact’ it denotes causation.  The methodology and data you cannot imply that one causes another.  So, the independent and dependent variables need to be addressed.

An issue also occurs with the sample and use of a proxy of current market participation to indicate past market participation.  The introduction claims that NZ has a high stock market participation, which does not match with the data.  The mean of in the sample is 13% participate in the stock market.  The paper by Giannetti and Koskinen (2010) has participation at 31%.  Your assumption that retired person current participation in the stock market indicates past participation is not borne out by your data.  You have lost 18% of the participation if these figures are accurate. 

This proxy problem also indicates that your measure of stock market participation may have bias.  That is, only people wealthy enough to own stocks pre-retirement and post retirement are included in your measure.  Those who owned stocks pre-retirement and then ceased to own stocks post retirement are not in your measure.  Yet, this group needs to be captured to address your research questions.  This issue is not trivial.  If I am facing retirement with a limited amount of wealth to generate income, then I will be risk adverse and take bonds or annuities to maintain income and wealth at an adequate level.  Conversely, if a person is facing retirement with excess wealth, then they have the capacity to take on more risk and participate in the stock market (perhaps whilst also buying bonds and annuities).  The measure would have captured the latter group only, and this would be amplifying your results. Can you address this?

Comments on the paper:

The paper should outline the three-pillar system briefly in the introduction so readers without an understanding can comprehend it the argument.  

Page 10- Outlines ‘the possible endogeneity issue’.  It is not clear what this issue is that you are trying to address.  Please outline what endogeneity issue you are addressing. 

The dependent variable should not be sufficiency.  This term implies it is a dummy variable which is either sufficient or not.  E.g.  have sufficient water to drink or I don’t.  

Please report the adjusted R2 statistics.  If you want to show the impact of your variable, you could report changes in the adjusted R2 when it is included. 

Use of insurance purchase as risk aversion is not clear to me.  It is confounded by home ownership and not a clear indication of an individual’s attitude towards financial risk taking.  

Round 2

Reviewer 1 Report

Hello Authors,

You have made significant revisions to the manuscript and the aim and scope of the research is apt.  I believe that you have an interesting idea that engagement with stock market participation in retirement should be encouraged.  Often models of retirement income assume that at retirement date, all wealth should be transferred to low risk assets.  However, with people living 25-35 years in retirement, this logic is incorrect. Your paper shows empirical evidence to support stock market participation.   

The paper presents an update on the endogeneity issue which I outlined in my earlier review.  This update is sufficient given the dataset you are working with.  

The paper is well written and I can follow the arguments presented.  I start to wonder about how much stock market participation would be ideal for retirees (e.g. 20% or 60% of their portfolio) and how this should change with age.  These are useful questions to consider but are the nature of future research and not required in this paper.

Overall, I believe this paper is good for publication.